# Point Transformer V2: Grouped Vector Attention and Partition-based Pooling

**Xiaoyang Wu**[1]    **Yixing Lao**[2]    **Li Jiang**[3]    **Xihui Liu**[1]    **Hengshuang Zhao**[1*]

[1]The University of Hong Kong    [2]Intel Labs    [3]Max Planck Institute

{xywu3, hszhao}@cs.hku.hk

## Abstract

As a pioneering work exploring transformer architecture for 3D point cloud understanding, Point Transformer achieves impressive results on multiple highly competitive benchmarks. In this work, we analyze the limitations of the Point Transformer and propose our powerful and efficient Point Transformer V2 model with novel designs that overcome the limitations of previous work. In particular, we first propose group vector attention, which is more effective than the previous version of vector attention. Inheriting the advantages of both learnable weight encoding and multi-head attention, we present a highly effective implementation of grouped vector attention with a novel grouped weight encoding layer. We also strengthen the position information for attention by an additional position encoding multiplier. Furthermore, we design novel and lightweight partition-based pooling methods which enable better spatial alignment and more efficient sampling. Extensive experiments show that our model achieves better performance than its predecessor and achieves state-of-the-art on several challenging 3D point cloud understanding benchmarks, including 3D point cloud segmentation on ScanNet v2 and S3DIS and 3D point cloud classification on ModelNet40. *Our code will be available at* https://github.com/Gofinge/PointTransformerV2.

## 1 Introduction

Point Transformer (PTv1) [1] introduces the self-attention networks to 3D point cloud understanding. Combining the vector attention [2] with a U-Net style encoder-decoder framework, PTv1 achieves remarkable performance in several 3D point cloud recognition tasks, including shape classification, object part segmentation, and semantic scene segmentation.

In this work, we analyze the limitations of Point Transformer (PTv1) [1] and propose a new elegant and powerful backbone named Point Transformer V2 (PTv2). Our PTv2 improves upon PTv1 with several novel designs, including the advanced grouped vector attention with improved position encoding, and the efficient partition-based pooling scheme.

The vector attention layers in PTv1 utilize MLPs as the weight encoding to map the subtraction relation of query and key into an attention weight vector that can modulate the individual channels of the value vector. However, as the model goes deeper and the number of channels increases, the number of weight encoding parameters also increases drastically, leading to severe overfitting and limiting the model depth. To address this problem, we present grouped vector attention with a more parameter-efficient formulation, where the vector attention is divided into groups with shared vector attention weights. Meanwhile, we show that the well-known multi-head attention [3] and the vector attention [2, 1] are degenerate cases of our proposed grouped vector attention. Our proposed grouped

---

*Corresponding Author

36th Conference on Neural Information Processing Systems (NeurIPS 2022).

vector attention inherits the merits of both vector attention and multi-head attention while being more powerful and efficient.

Furthermore, point positions provide important geometric information for 3D semantic understanding. Hence, the positional relationship among 3D points is more critical than 2D pixels. However, previous 3D position encoding schemes mostly follow the 2D ones and do not fully exploit the geometric knowledge in 3D coordinates. To this end, we strengthen the position encoding mechanism by applying an additional position encoding multiplier to the relation vector. Such a design strengthens the positional relationship information in the model, and we validate its effectiveness in our experiments.

Moreover, it is worth noting that the irregular, non-uniform spatial distributions of points are significant challenges to the pooling modules for point cloud processing. Previous point cloud pooling approaches rely on a combination of sampling methods (e.g. farthest point sampling [4] or grid sampling [5]) and neighbor query methods (e.g. kNN or radius query), which is time-consuming and not spatially well-aligned. To overcome this problem, we go beyond the pooling paradigm of combining sampling and query, and divide the point cloud into non-overlapping partitions to directly fuse points within the same partition. We use uniform grids as partition divider and achieve significant improvement.

In conclusion, we propose Point Transformer V2, which improves Point Transformer [1] from several perspectives:

- We propose an effective grouped vector attention (GVA) with a novel weight encoding layer that enables efficient information exchange within and among attention groups.
- We introduce an improved position encoding scheme to utilize point cloud coordinates better and further enhance the spatial reasoning ability of the model.
- We design the partition-based pooling strategy to enable more efficient and spatially better-aligned information aggregation compared to previous methods.

We conducted extensive analysis and controlled experiments to validate our designs. Our results indicate that PTv2 outperforms predecessor works and sets the new state-of-the-art on various 3D understanding tasks.

## 2  Related Works

**Image transformers.** With the great success of ViT [6], the absolute dominance of convolution in vision tasks is shaken by Vision Transformer, which becomes a trend in 2D image understanding [7, 8, 9, 10]. ViT introduces the far-reaching scaled dot-product self-attention and multi-head self-attention theory [3] in NLP into vision by considering image patches as tokens. However, operating global attention on the entire image consumes excessive memory. To solve the memory consumption problem, Swin Transformer [7] introduces the grid-based local attention mechanism to operate the transformer block in a sequence of shifted windows.

**Point cloud understanding.** Learning-based methods for processing 3D point clouds can be classified into the following types: projection-based, voxel-based, and point-based networks. An intuitive way to process irregular inputs like point clouds is to transform irregular representations into regular ones. Projection-based methods project 3D point clouds into various image planes and utilize 2D CNN-based backbones to extract feature representations [11, 12, 13, 14]. An alternative approach operates convolutions in 3D by transforming irregular point clouds into regular voxel representations [15, 16]. Those voxel-based methods suffer from inefficiency because of the sparsity of point clouds until the introduction and implementation of sparse convolution [17, 18]. Point-based methods extract features directly from the point cloud rather than projecting or quantizing irregular point clouds onto regular grids in 2D or 3D [19, 4, 20, 5]. The recently proposed transformer-based point cloud understanding approaches, introduced in the next paragraph, are also categorized into point-based methods.

**Point cloud transformers.** Transformer-based networks belong to the category of point-based networks for point cloud understanding. During the research upsurge of vision transformers, at almost the same period, Zhao et al. [1] and Guo et al. [21] published their explorations of applying attention to point cloud understanding, becoming pioneers in this direction. The PCT [21] proposed by Guo et al. performs global attention directly on the point cloud. Their work, similar to ViT, is limited

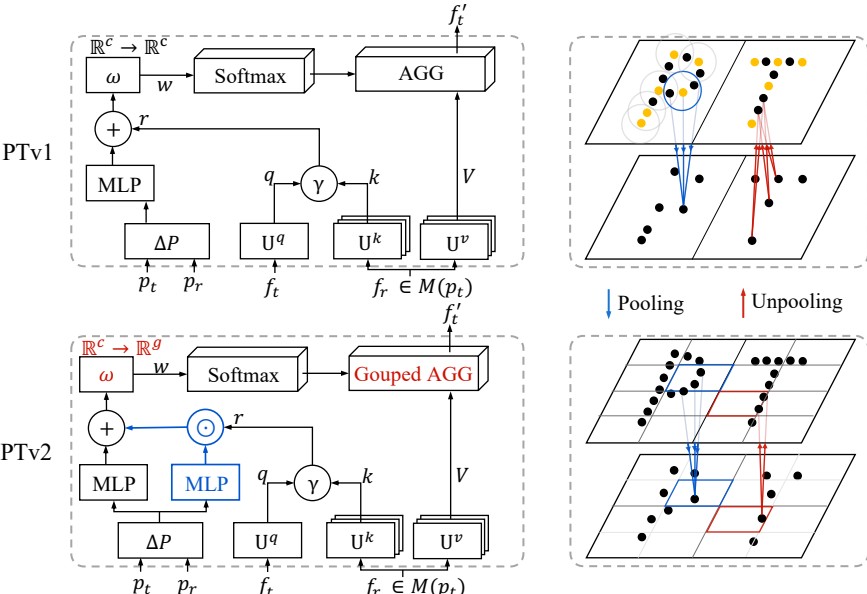

Figure 1: Comparison of the attention, position encoding, and pooling mechanisms between PTv1 and PTv2. Top-left: the vector attention (Sec. 3.1) with position encoding (Sec. 3.3) in PTv1 . Bottom-left: our grouped vector attention (Sec. 3.2, denoted by red) with improved position encoding (Sec. 3.3, denoted by blue) in PTv2. Top-right: the sampling-based pooling and interpolation-based unpooling in PTv1. Bottom-right: our partition-based pooling and unpooling in PTv2 (Sec. 3.4).

by memory consumption and computational complexity. Meanwhile, based on the vector attention theory proposed in SAN [2], Point Transformer [1] proposed by Zhao et al. directly performs local attention between each point and its adjacent points, which alleviated the memory problem mentioned above. Point Transformer achieves remarkable results in multiple point cloud understanding tasks and state-of-art results for several competitive challenges. In this work, we analyze the limitations of the Point Transformer [1], and propose several novel architecture designs for the attention and pooling module, to improve the effectiveness and efficiency of the Point Transformer. Our proposed model, Point Transformer V2, performs better than the Point Transformer across a variety of 3D scene understating tasks.

## 3   Point Transformer V2

We analyze the limitations of Point Transformer V1 (PTv1) [1] and propose our Point Transformer V2 (PTv2), including several improved modules upon PTv1. We begin by introducing the mathematical formulations and revisiting the vector self-attention used in PTv1 in Sec. 3.1. Based on the observation that the parameters of PTv1 increases drastically with the increased model depth and channel size, we propose our powerful and efficient grouped vector attention in Sec. 3.2. Further, we introduce our improved position encoding in Sec. 3.3 and the new pooling method in Sec. 3.4. We finally describe our network architecture in Sec. 3.5.

### 3.1   Problem Formulation and Background

**Problem formulation.** Let $\mathcal{M} = (\mathcal{P}, \mathcal{F})$ be a 3D point cloud scene containing a set of points $\boldsymbol{x}_i = (\boldsymbol{p}_i, \boldsymbol{f}_i) \in \mathcal{M}$, where $\boldsymbol{p}_i \in \mathbb{R}^3$ represents the point position, and $\boldsymbol{f}_i \in \mathbb{R}^c$ represents the point features. Point cloud semantic segmentation aims to predict a class label for each point $\boldsymbol{x}_i$, and the goal of scene classification is to predict a class label for each scene $\mathcal{M}$. $\mathcal{M}(\boldsymbol{p})$ denotes a mapping function that maps the point at position $\boldsymbol{p}$ to a subset of $\mathcal{M}$ denoted as "reference set". Next, we revisit the self-attention mechanism used in PTv1 [1].

**Local attention.** Conducting the global attention [6, 21] over all points in a scene is computationally heavy and infeasible for large-scale 3D scenes. Therefore, we apply local attention where the attention for each point $x_i$ works within a subset of points, i.e., reference point set, $\mathcal{M}(p_i)$.

*Shifted-grid attention* [7], where attention is alternatively applied over two sets of non-overlapping image grids, has become is a common practice [22, 23, 24, 25] for image transformers. Similarly, the 3D space can be split into uniform non-overlapping grid cells, and the reference set is defined as the points within the same grid, i.e., $\mathcal{M}(p_i) = \{(p_j, f_j) \mid p_j$ in the same grid cell as $p_i\}$. However, such attention relies on a cumbersome shift grid operation to achieve a global receptive field, and it does not work well on point clouds where the point densities within different grids are not consistent.

PTv1 adopts *neighborhood attention*, where the reference point set is a local neighborhood of the given point, i.e., $\mathcal{M}(p_i) = \{(p_j, f_j) \mid p_j \in \text{Neighborhood}(p_i)\}$. Specifically, the neighborhood point set $\mathcal{M}(p_i)$ is defined as the $k$ nearest neighboring (kNN) points of $p_i$ in PTv1. Our experiments (Sec. 4.3) show that neighborhood attention is more effective than shifted-grid attention, so our approach adopts the neighborhood attention.

**Scalar attention and vector attention.** Given a point $x_i = (p_i, f_i) \in \mathcal{M}$, we apply linear projections or MLPs to project the point features $f_i$ to the feature vectors of query $q_i$, key $k_i$, and value $v_i$, each with $c_h$ channels. The standard *scalar attention (SA)* operated on the point $x_i$ and its reference point set $\mathcal{M}(p_i)$ can be represented as follows,

$$w_{ij} = \langle q_i, k_j \rangle / \sqrt{c_h}, \qquad f_i^{\text{attn}} = \sum_{x_j \in \mathcal{M}(p_i)} \text{Softmax}(w_i)_j v_j, \qquad (1)$$

The attention weights in the above formulation are scalars computed from the scaled dot-product [3] between the query and key vectors. Multi-head scalar attention (MSA) [3] is an extension of SA which runs several scalar attentions in parallel. MSA is widely applied in transformers, and we will show in Sec. 3.2 that MSA is a degenerate case of our proposed grouped vector attention.

Instead of the scalar attention weights, PTv1 applies *vector attention*, where the attention weights are vectors that can modulate the individual feature channels. In SA, the scalar attention is computed by the scaled dot-product between the query and key vectors. In vector attention, a weight encoding function encodes the relation between query and key to a vector. The vector attention [2] is formulated as follows,

$$w_{ij} = \omega(\gamma(q_i, k_j)), \qquad f_i^{\text{attn}} = \sum_{x_j \in \mathcal{M}(p_i)} \text{Softmax}(W_i)_j \odot v_j, \qquad (2)$$

where $\odot$ is the Hadamard product. $\gamma$ is a relation function (e.g., subtraction). $\omega : \mathbb{R}^c \mapsto \mathbb{R}^c$ is a learnable weight encoding (e.g., MLP) that computes the attention vectors to re-weight $v_j$ by channels before aggregation. Fig. 2 (a) shows a method using vector attention with linear weight encoding.

## 3.2 Grouped Vector Attention

In vector attention, as the network goes deeper and there are more feature encoding channels, the number of parameters for the weight encoding layer increases drastically. The large parameter size restricts the efficiency and generalization ability of the model. In order to overcome the limitations of vector attention, we introduce the grouped vector attention, as illustrated in Fig. 1 (left).

**Attention groups.** We divide channels of the value vector $v \in \mathbb{R}^c$ evenly into $g$ groups ($1 \le g \le c$). The weight encoding layer outputs a grouped attention vector with $g$ channels instead of $c$ channels. Channels of $v$ within the same attention group share the same scalar attention weight from the grouped attention vector. Mathematically,

$$w_{ij} = \omega(\gamma(q_i, k_j)), \qquad f_i^{attn} = \sum_{x_j}^{\mathcal{M}(p_i)} \sum_{l=1}^{g} \sum_{m=1}^{c/g} \text{Softmax}(W_i)_{jl} v_j^{lc/g+m}, \qquad (3)$$

where $\gamma$ is the relation function and $\omega : \mathbb{R}^c \mapsto \mathbb{R}^g$ is the learnable *grouped weight encoding* defined in the next paragraph. The second equation in Eq. 3 is the *grouped vector aggregation*. Fig. 2 (a) presents a vanilla GVA implemented by a fully connected weight encoding, the number of the grouped weight encoding function parameters reduced compared with the vector attention (Fig. 2 (b)), leading to a more powerful and efficient model.

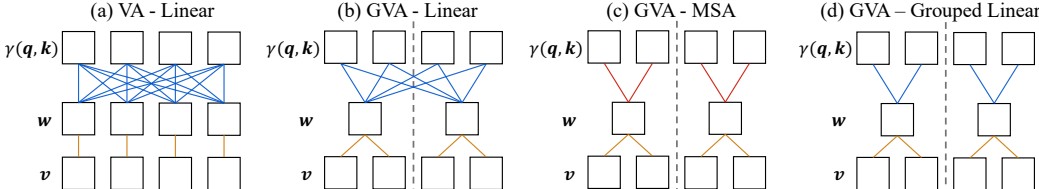

Figure 2: Comparison of various weight encoding functions. Each square represents a scalar, and each row of them represents a vector. The three rows represent relation vector, weight vector, and value vector from top to bottom. The attention groups are separated by dash lines. For demonstration, we assume the feature dimension is 4 and the number of attention groups (applicable to b, c, d) is 2. Lines with different colors refer to different operations, blue lines represent learnable parameters act on input relation scalar, while red lines represent multiply by the input relation scalar. Orange lines identify which value feature is affected by the input scalar weight.

**GVA is a generalized formulation of VA and MSA.** Our GVA degenerates to vector attention (VA) when $g = c$, and it degenerates to multi-head self-attention (MSA) if $\omega$ in Eq. 3 is defined as follows,

$$\omega(\boldsymbol{r}) = \boldsymbol{r} \underbrace{\begin{bmatrix} \mathbf{1}_{1 \times c_g} & \mathbf{0}_{1 \times c_g} & \cdots & \mathbf{0}_{1 \times c_g} \\ \mathbf{0}_{1 \times c_g} & \mathbf{1}_{1 \times c_g} & \cdots & \mathbf{0}_{1 \times c_g} \\ \vdots & \vdots & \ddots & \vdots \\ \mathbf{0}_{1 \times c_g} & \mathbf{0}_{1 \times c_g} & \cdots & \mathbf{1}_{1 \times c_g} \end{bmatrix}^{T}}_{g \times c_g} \frac{1}{\sqrt{c_g}}, \tag{4}$$

where $c_g = c/g$ and $r \in \mathbb{R}^{1 \times c}$.

**Grouped linear.** Inspired by the weight encoding function of MSA, we design the grouped linear layer $\zeta(r) : \mathbb{R}^c \mapsto \mathbb{R}^g$ where different groups of the input vector are projected with different parameters independently. Grouped linear further reduce the number of parameters in the weight encoding function. Our final adopted grouped weight encoding function is composed of the grouped linear layer, normalization layer, activation layer, and a fully connected layer to allow inter-group information exchange. Mathematically,

$$\zeta(\boldsymbol{r}) = \boldsymbol{r} \underbrace{\begin{bmatrix} \boldsymbol{p}_1 & \mathbf{0}_{1 \times c_g} & \cdots & \mathbf{0}_{1 \times c_g} \\ \mathbf{0}_{1 \times c_g} & \boldsymbol{p}_2 & \cdots & \mathbf{0}_{1 \times c_g} \\ \vdots & \vdots & \ddots & \vdots \\ \mathbf{0}_{1 \times c_g} & \mathbf{0}_{1 \times c_g} & \cdots & \boldsymbol{p}_g \end{bmatrix}^{T}}_{g \times c_g}, \tag{5}$$

$$\omega(\boldsymbol{r}) = \text{Linear} \circ \text{Act} \circ \text{Norm}(\zeta(\boldsymbol{r})), \tag{6}$$

where $c_g = c/g$, $\boldsymbol{p}_1, \ldots, \boldsymbol{p}_g \in \mathbb{R}_g^c$ are learnable parameters, and $\circ$ represents function composition.

### 3.3 Position Encoding Multipler

Different from the discrete, regular-grid pixels in 2D images, points in the 3D point cloud are unevenly distributed in a continuous Euclidean Metric space, making the spatial relationship in 3D point cloud much more complicated than 2D images. In transformers and attention modules, the spatial information is obtained with the position encoding $\delta_{bias}(\boldsymbol{p}_i - \boldsymbol{p}_j)$ added to the relation vector $\gamma(\boldsymbol{q}_i, \boldsymbol{k}_j)$ as a bias.

Due to the generalization limitation of vector attention in PTv1 mentioned in Sec. 3.2, adding more position encoding capacity to vector attention will not help to improve the performance. In PTv2, the grouped vector attention has an effect of reducing overfitting and enhancing generalization. With grouped vector attention restricting the capacity of the attention mechanism, we strengthen the position encoding with an additional multiplier $\delta_{mul}(\boldsymbol{p}_i - \boldsymbol{p}_j)$ to the relation vector, which focuses on learning complex point cloud positional relations. As shown in Fig. 1 (left), our improved position

encoding is as follows,

$$\boldsymbol{w}_{ij} = \omega(\delta_{mul}(\boldsymbol{p}_i - \boldsymbol{p}_j) \odot \gamma(\boldsymbol{q}_i, \boldsymbol{k}_j) + \delta_{bias}(\boldsymbol{p}_i - \boldsymbol{p}_j)), \tag{7}$$

where $\odot$ is the Hadamard product. $\delta_{mul}, \delta_{bias} : \mathbb{R}^d \mapsto \mathbb{R}^d$ are two MLP position encoding functions, which take relative positions as input. Position encoding multiplier compliments group vector attention to achieve a good balance of network capacity.

## 3.4 Partition-based Pooling

Traditional sampling-based pooling procedures adopted by other point-based methods use a combination of sampling and query methods. In the sampling stage, farthest point sampling [4] or grid sampling [5] is used to sample points reserved for the following encoding stage. For each sampled point, a neighbor query is performed to aggregate information from the neighboring points. In these sampling-based pooling procedures, the query sets of points are not spatially-aligned since the information density and overlap among each query set are not controllable. To address the problem, we propose a more efficient and effective partition-based pooling approach, as shown in Fig. 1.

**Pooling.** Given a point set $\mathcal{M} = (\mathcal{P}, \mathcal{F})$, we partition $\mathcal{M}$ into subsets $[\mathcal{M}_1, \mathcal{M}_2, ..., \mathcal{M}_{n'}]$ by separating the space into non-overlapping partitions. We fusion each subset of points $\mathcal{M}_i = (\mathcal{P}_i, \mathcal{F}_i)$ from a single partition as follows,

$$\boldsymbol{f}'_i = \text{MaxPool}(\{\boldsymbol{f}_j \boldsymbol{U} \mid \boldsymbol{f}_j \in \mathcal{F}_i\}), \qquad \boldsymbol{p}'_i = \text{MeanPool}(\{\boldsymbol{p}_j \mid \boldsymbol{p}_j \in \mathcal{P}_i\}), \tag{8}$$

where $(\boldsymbol{p}'_i, \boldsymbol{f}'_i)$ is the position and features of pooling point aggregated form subset $\mathcal{M}_i$, and $\boldsymbol{U} \in \mathbb{R}^{c \times c'}$ is the linear projection. Collecting the pooling points from $n'$ subsets gives us the point set $\mathcal{M}' = \{\boldsymbol{p}'_i, \boldsymbol{f}'_i\}_{i=1}^{n'}$ for the next stage of encoding. In our implementation, we use uniform grids to partition the point cloud space, and thus our partition-based pooling is also called grid pooling.

**Unpooling.** The common practice of unpooling by interpolation is also applicable to partition-based pooling. Here we introduce a more straightforward and efficient unpooling method. To unpool the fused point set $\mathcal{M}'$ back to $\mathcal{M}$, the point locations in $\mathcal{M}$ are record from the pooling process, and we only need to obtain the features for each point in $\mathcal{M}$. With the help of the grid-based partitioning $[\mathcal{M}_1, \mathcal{M}_2, ..., \mathcal{M}_{n'}]$ during the pooling stage, we can map point feature to all points from the same subset,

$$\boldsymbol{f}_i^{up} = \boldsymbol{f}'_j, \qquad \text{if } (\boldsymbol{p}_i, \boldsymbol{f}_i) \in \mathcal{M}_j. \tag{9}$$

## 3.5 Network Architecture

**Backbone structure.** Following previous works [18, 1], we adopt the U-Net architecture with skip connections. There are four stages of encoders and decoders with block depths [2, 2, 6, 2] and [1, 1, 1, 1], respectively. The grid size multipliers for the four stages are [x3.0, x2.5, x2.5, x2.5], representing the expansion ratio over the previous pooling stage. The attention is conducted in a local neighborhood, described in "neighborhood attention" in Sec. 3.1. In Sec. 4.3 we compare the neighborhood attention with shift-grid attention.

The initial feature dimension is 48, and we first embed the input channels to this number with a basic block with attention groups of 6. Then, we double this feature dimension and attention groups each time entering the next encoding stage. For the four encoding stages, the feature dimensions are [96, 192, 384, 384], and the corresponding attention groups are [12, 24, 48, 48].

**Output head.** For point cloud semantic segmentation, we apply an MLP to map point features produced by the backbone to the final logits for each point in the input point set. For point cloud classification, we apply global average pooling over the point features produced by the encoding stages to obtain a global feature vector, followed by an MLP classifier for prediction.

# 4 Experiments

To validate the effectiveness of the proposed method, we conduct experimental evaluations on ScanNet v2 [44] and S3DIS [45] for semantic segmentation, and ModelNet40 [46] for shape classification. Implementation details are available in the appendix.

Table 1: Semantic segmentation on ScanNet v2.

| Method | Input | Val | Test |
|---|---|---|---|
| PointNet++ [4] | point | 53.5 | 55.7 |
| 3DMV [26] | point | - | 48.4 |
| PanopticFusion [27] | point | - | 52.9 |
| PointCNN [28] | point | - | 45.8 |
| PointConv [29] | point | 61.0 | 66.6 |
| JointPointBased [30] | point | 69.2 | 63.4 |
| PointASNL [31] | point | 63.5 | 66.6 |
| SegGCN [32] | point | - | 58.9 |
| RandLA-Net [33] | point | - | 64.5 |
| KPConv [5] | point | 69.2 | 68.6 |
| JSENet [34] | point | - | 69.9 |
| FusionNet [35] | point | - | 68.8 |
| SparseConvNet [17] | voxel | 69.3 | 72.5 |
| MinkUNet [18] | voxel | 72.2 | 73.6 |
| PTv1 [1] | point | 70.6 | - |
| PTv2 (ours) | point | **75.4** | **75.2** |

Table 2: Semantic segmentation on S3DIS Area 5.

| Method | Input | OA | mAcc | mIoU |
|---|---|---|---|---|
| PointNet [19] | point | - | 49.0 | 41.1 |
| SegCloud [36] | point | - | 57.4 | 48.9 |
| TanConv [37] | point | - | 62.2 | 52.6 |
| PointCNN [28] | point | 85.9 | 63.9 | 57.3 |
| PointWeb [20] | point | 87.0 | 66.6 | 60.3 |
| HPEIN [38] | point | 87.2 | 68.3 | 61.9 |
| GACNet [39] | point | 87.8 | - | 62.9 |
| PAT [40] | point | - | 70.8 | 60.1 |
| ParamConv [41] | point | - | 67.0 | 58.3 |
| SPGraph [42] | point | 86.4 | 66.5 | 58.0 |
| SegGCN [32] | point | 88.2 | 70.4 | 63.6 |
| MinkUNet [18] | voxel | - | 71.7 | 65.4 |
| PAConv [43] | point | - | - | 66.6 |
| KPConv [5] | point | - | 72.8 | 67.1 |
| PTv1 [1] | point | 90.8 | 76.5 | 70.4 |
| PTv2 (ours) | point | **91.1** | **77.9** | **71.6** |

## 4.1 Semantic Segmentation

**Data and metric.** For semantic segmentation, we experiment on ScanNet v2 [44] and S3DIS [45]. The ScanNet v2 dataset contains 1,513 room scans reconstructed from RGB-D frames. The dataset is divided into 1,201 scenes for training and 312 for validation. Point clouds for the model input are sampled from vertices of reconstructed meshes, and each sampled point is assigned a semantic label from 20 categories (wall, floor, table, etc.). The S3DIS dataset for semantic scene parsing consists of 271 rooms in six areas from three different buildings. Following a common protocol [36, 4, 1], area 5 is withheld during training and used for testing. Different from ScanNet v2, points of S3DIS are densely sampled on the mesh surfaces and annotated into 13 categories. Following a standard protocol [4], we use mean class-wise intersection over union (mIoU) as the evaluation metric for validation and test set of ScanNet v2. And we use mean class-wise intersection over union (mIoU), mean of class-wise accuracy (mAcc), and overall point-wise accuracy (OA) for evaluating performance on S3DIS area5.

**Performance comparison.** Table 1 and Table 2 show the results of our PTv2 model compared with previous methods on ScanNet v2 and S3DIS, respectively. Our PTv2 model outperforms prior methods in all evaluation metrics. Notably, PTv2 significantly outperforms PTv1 [1] by 4.8% mIoU on the ScanNet v2 validation set.

**Visualization.** The qualitative results of point cloud semantic segmentation are shown in Fig. 3 and Fig. 4. Our PTv2 model is able to predict semantic segmentation results that are quite close the ground-truth. It is worth noting that our model can capture the detailed structure information and predict the correct semantics for challenging scenarios. For example, in the S3DIS scenes with chairs, PTv2 is able to cleanly predict the chair legs and armrests.

## 4.2 Shape Classification

**Data and metric.** We test our proposed PTv2 model for 3D point cloud classification on ModelNet40 dataset. The ModelNet40 [46] dataset consists of 12,311 CAD models belonging to 40 object categories. 9,843 models are split out for training, and the rest 2,468 models are reserved for testing. Following the common practice in the community, we report the class-average accuracy (mAcc) and overall accuracy (OA) on the test set.

**Performance comparison.** We test our PTv2 model and compare it with previous models on the ModelNet40 dataset for shape classification. Results are shown in Table 3, demonstrating that our proposed PTv2 model achieves state-of-the-art performance on ModelNet40 shape classification.

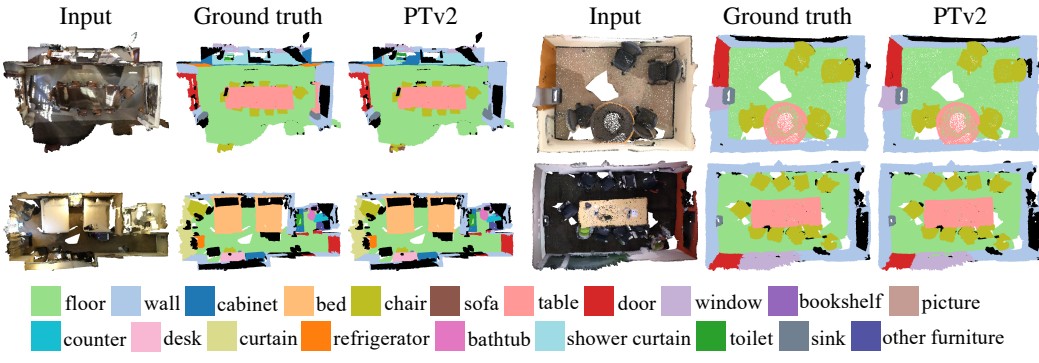

floor | wall | cabinet | bed | chair | sofa | table | door | window | bookshelf | picture
counter | desk | curtain | refrigerator | bathtub | shower curtain | toilet | sink | other furniture

Figure 3: Visualization of semantic segmentation results on ScanNet v2.

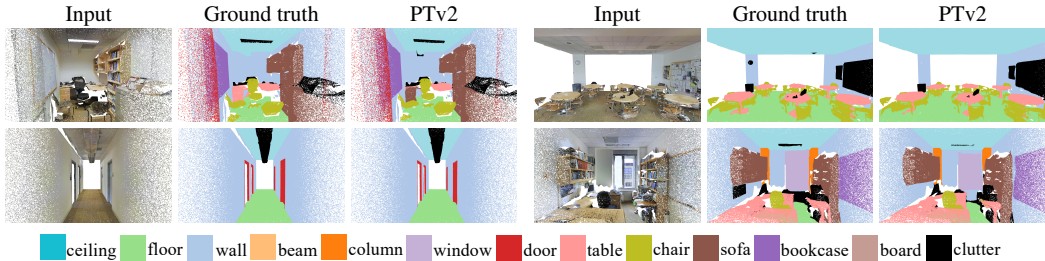

ceiling | floor | wall | beam | column | window | door | table | chair | sofa | bookcase | board | clutter

Figure 4: Visualization of semantic segmentation results on S3DIS.

## 4.3 Ablation Study

We conduct ablation studies to examine the effectiveness of each module in our design. The ablation study results are reported on ScanNet v2 validation set.

**Attention type.** We first investigate the effects of different attention designs. We experiment with two types of local attention introduced in Sec. 3.1, namely shifted-grid attention and neighborhood attention [1]. Then, to validate the effectiveness of our proposed grouped vector attention (denoted as "GVA"), we compare it with the commonly-used multi-head self-attention (denoted as "MSA"). We use the vanilla position encoding in PTv1 [1] and our proposed partition-based pooling scheme in all of the experiments in Table 4. It shows neighborhood attention performs significantly better than shifted-grid attention, indicating that the neighborhood attention is better suited for point clouds which are non-uniformly distributed. Moreover, our proposed grouped vector attention consistently outperforms the commonly-used multi-head self-attention with both shifted-grid attention and neighborhood attention. So our grouped vector attention is not only more efficient, but also more effective, than multi-head self-attention. The comparison between GVA and MSA indicates the effectiveness of the learnable parameters in the grouped linear layer of the grouped weight encoding in Sec. 3.2.

**Weight encoding.** We study the effects of different weight encoding functions $\omega$ in Table 6. The weight encoding functions are introduced in Sec. 3.1 and Sec. 3.2, and different attention mechanisms adopt different weight encoding functions. We use the vanilla position encoding in PTv1 [1] and our proposed grid pooling scheme in all of the experiments in Table 6. We experimented with the following weight encoding functions: (1) The weight encoding for multi-head scalar attention in Eq. 4, denoted as "MSA". (2) Weight encoding as a linear layer denoted as "L". (3) The grouped linear layer, which is $\zeta$ in Eq. 5, denoted as "GL". (4) The linear layer followed by batch normalization, activation, and another linear layer, denoted as "L+N+A+L". (5) The grouped linear layer, followed by batch normalization, activation, and a linear layer, denoted as "GL+N+A+L". (5) is also the grouped weight encoding function used for our grouped vector attention, introduced as $\omega$ in Eq. 5. Results in Table 6 demonstrate that our grouped weight encoding function outperforms other compared designs. Specifically, comparing (1), (3) and (5), GL slightly outperforms MSA but adding additional inter-group information exchange combined with proper normalization and activation can boost the performance to be better than MSA. Moreover, the comparison between (5) and (4) and the comparison between (3) and (2) both indicate that our grouped linear layer outperforms the naive linear layer, even though the grouped linear layer has $g$ times fewer parameters and requires less computing than the linear layer.

Table 3: Shape classification on ModelNet40.

| Method | mAcc (%) | OA (%) |
|---|---|---|
| PointNet [19] | 86.0 | 89.2 |
| PointNet++ [4] | - | 91.9 |
| PointCNN [28] | 88.1 | 92.5 |
| PointConv [29] | - | 92.5 |
| KPConv [5] | - | 92.9 |
| DGCNN [47] | 90.2 | 92.9 |
| RS-CNN [48] | - | 92.9 |
| PointASNL [31] | - | 92.9 |
| DensePoint [49] | - | 93.2 |
| PosPool [50] | - | 93.2 |
| GBNet [51] | 91.0 | 93.8 |
| PCT [21] | - | 93.2 |
| PA-DGC [43] | - | 93.9 |
| CurveNet [52] | - | **94.2** |
| PTv1 [1] | 90.6 | 93.7 |
| PTv2 (ours) | **91.6** | **94.2** |

Table 4: Attention type ablation.

| Local Type | Mechanism Type | mIoU (%) |
|---|---|---|
| Shifted-Grid | MSA | 71.6 |
| | GVA | 72.5 |
| **Neighborhood** | MSA | 73.9 |
| | **GVA** | **75.0** |

Table 5: Pooling method ablation.

| Pooling Method | Pooling Ratio | Grid Size Multipliers | mIoU (%) |
|---|---|---|---|
| FPS | 1/4 | - | 74.4 |
| | 1/6 | - | 72.9 |
| **Grid** | ~1/4 | [×3.0, ×2.0, ×2.0, ×2.0] | 75.2 |
| | ~1/4 | [×4.0, ×2.0, ×2.0, ×2.0] | 75.0 |
| | ~1/6 | [×3.0, ×2.5, ×2.5, ×2.5] | **75.4** |
| | ~1/6 | [×4.0, ×2.5, ×2.5, ×2.5] | 74.7 |

Table 6: Weight encoding ablation.

| ID | Weight encoding | mIoU (%) |
|---|---|---|
| (1) | MSA | 73.9 |
| (2) | L | 73.8 |
| (3) | GL | 74.1 |
| (4) | L+N+A+L | 74.7 |
| (5) | **GL+N+A+L (ours)** | **75.0** |

Table 7: Module design ablation.

| ID | GVA | PE Mul | Grid Pool | Map Unpool | mIoU (%) |
|---|---|---|---|---|---|
| I | | | | | 72.3 |
| II | ✓ | | | | 73.8 |
| III | ✓ | ✓ | | | 74.4 |
| IV | ✓ | ✓ | ✓ | | 74.9 |
| V | ✓ | ✓ | ✓ | ✓ | **75.4** |

**Pooling methods.** In Sec. 3.4 we discuss the potential limitations of the sampling-based pooling in PTv1 and propose a new pooling and unpooling scheme based on non-overlapping partitions. We also name a simple and effective grid-based implement of our partition-based pooling as grid pooling. To further examine the superiority of our method, we experiment with different pooling-unpooling schemes in Table 5.

For our partition-based pooling implemented by a grid, the base grid size is 0.02 meters, which is identical to the voxelization grid size during data pre-processing. The grid size multipliers are the grid size expansion ratio over the previous pooling stage. For example, $[\times 4.0, \times 2.0, \times 2.0, \times 2.0]$ means that the grid sizes are: $[0.08, 0.16, 0.32, 0.64]$ meters, respectively. We choose a relatively large value for initial grid sizes ($\times 3.0$ and $\times 4.0$) to provide sufficiently large receptive fields, which is analogous to the common practice in image transformers [6]. For subsequent pooling stages, we observe that $\times 2.0$ grid size increase results in an approximate pooling ratio of 4 for the point cloud, while $\times 2.5$ grid size increase results in an approximate pooling ratio of 6. We choose the same sampling ratio of 4 and 6 for sampling-based pooling to ensure a fair comparison.

The results in Table 5 illustrate that our partition-based pooling achieves higher mIoU than the sampling-based method. For sampling-based pooling with farthest point sampling, the performance decreases significantly when the sampling ratio increases from 4 to 6. However, for our partition-based pooling implemented by grid, we observe that initial grid size and subsequent grid size multipliers do not significantly affect the overall performance, so we can use larger grid sizes to reduce the number of points in each stage to save memory.

**Module design.** We ablate different modules introduced in our PTv2: grouped vector attention (VGA), position encoding multiplier (PE Mul), partition-based pooling implemented by grid (Grid Pool), and partition map unpooling (Map Unpool) and the results are illustrated in Table 7. The model adopts in Experiment I is PTv1 [1], which serves as a baseline result of our design. Benefiting from structural parameter adjustments and better data processing, which are also shared with the rest of the experiments, our baseline result increased from 70.6% to 72.3%. Experiment II to V add each of our proposed components in turns, gradually increasing our baseline result to 75.4%. The increasing mIOU indicates the effectiveness of each component.

Table 8: Model performance and amortized latency with different pooling methods.

| | FPS-kNN [4] | | Grid-kNN [5] | | Grid pooling (ours) | |
|---|---|---|---|---|---|---|
| Pooling Rate | 1/4 | 1/6 | ~1/4 | ~1/6 | ~1/4 | ~1/6 |
| Time (ms) | 1007 | 785 | 389 | 356 | 318 | **266** |
| mIoU (%) | 74.4 | 72.9 | 74.1 | 73.4 | 75.2 | **75.4** |

Table 9: Model parameters and amortized latency of several networks.

| | ① PTv1 | ② ① + GVA (L) | ③ ① + GVA (GL) | ④ ① + GVA (GL-N-A-L) | ⑤ ④ + GP | ⑥ ⑤ + PEM |
|---|---|---|---|---|---|---|
| Params (M) | 11.4 | 9.8 | 9.6 | 9.6 | 9.6 | 12.8 |
| Time (ms) | 1023 | 991 | 951 | 971 | **220** | 266 |
| mIoU (%) | 72.3 | 73.0 | 73.2 | 74.2 | 75.0 | **75.4** |

## 4.4 Model Complexity and Latency

We further conduct model complexity and latency studies to examine the superior efficiency of several design in our work. We record the amortized forward time for each scan in the ScanNet v2 validation set with batch size 4 on a single TITAN RTX.

**Pooling methods.** Table 8 shows the forward time and mIoU of PTv2 with different pooling methods and pooling ratios. We compare our pooling method with two classical sampling-based pooling methods: FPS-kNN and Grid-kNN. FPS-kNN pooling [4, 1] uses farthest point sampling (FPS) to sample a specified number of points and then query $k$ nearest neighbor points for pooling. We call the pooling method in Strided KPConv [5] Grid-kNN pooling, as it uses a uniform grid to sample points and then applies the kNN method to index neighbors. This leads to uncontrollable overlaps of the pooling receptive fields. As shown in the table, our grid pooling method is not only faster but also achieves higher mIoUs.

**Module design.** Table 9 summarizes comparisons of model complexity, time consumption, and evaluation performances on ScanNet v2 validation set. Meanwhile, we drop the first batch of forwarding time for GPU preparation. In Table 9, GVA refers to grouped vector attention. L refers to grouped weight encoding implemented by a single Linear. GL refers to grouped weight encoding implemented by a Grouped Linear. GL-N-A-L refers to the grouped linear layer, followed by batch normalization, activation, and a linear layer as grouped weight encoding function. GP refers to partition-based pooling implemented by grid. PEM refers to the position encoding multiplier. To ensure fair comparison, PTv1 is set to be the same depth and feature dimensions as our model architecture of PTv2. By comparing experiment ① and ②, we can study the effect of GVA. The same spirit goes on for experiment ③, ④, and ⑤, where each experiments adds one additional module, so that we can study the effect of the added module respectively.

Comparing experiments ①, ②, ③, and ④, the introduction of grouped vector attention (GVA) with grouped weight encoding dramatically improves the model performance and slightly reduces execution time. The comparison between ④ and ⑤ indicates that the grid pooling strategy can significantly speed up the network and further enhance the generalization ability of our model. Position encoding multiplier is the only design that increases the number of model parameters, but experiment ⑥ demonstrates its effectiveness in improving performance. Meanwhile, our model is still lightweight compared to voxel-based backbones, such as MinkUNet42 [18] with 37.9M parameters.

## 5 Conclusion

We propose Point Transformer V2 (PTv2), a powerful and efficient transformer-based backbone for 3D point cloud understanding. Our work makes several non-trivial improvements upon Point Transformer V1 [1], including the grouped vector attention, improved position encoding, and partition-based pooling. Our PTv2 model achieves state-of-the-art performance on point cloud classification and semantic segmentation benchmarks.

## Acknowledgements

This work is supported in part by HKU Startup Fund and HKU Seed Fund for Basic Research. We also appreciate the supporting of computing resources by SmartMore Corporation.

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
