# Point Transformer V2: Grouped Vector Attention and Improved Sampling – Supplementary Material

**Xiaoyang Wu**[1]  **Yixing Lao**[1]  **Li Jiang**[2]  **Xihui Liu**[1]  **Hengshuang Zhao**[1*]

[1]The University of Hong Kong  [2]Max Planck Institute

{xywu3, hszhao}@cs.hku.hk

In the appendix, we provide more experiment details in Sec. A, more experiment results in Sec. B.

## A  Experiment Details

This section describes the model architectures adopted in the experiments and describes the experimental settings for each dataset in detail.

### A.1  Model Architecture

In Fig. 1, we show the detailed network architectures for semantic segmentation and shape classification. Tuples under each stage block indicate the number of sampled points and feature dimensions of attention blocks, and the number of sampled points is determined by grid sizes, as specified in the main paper.

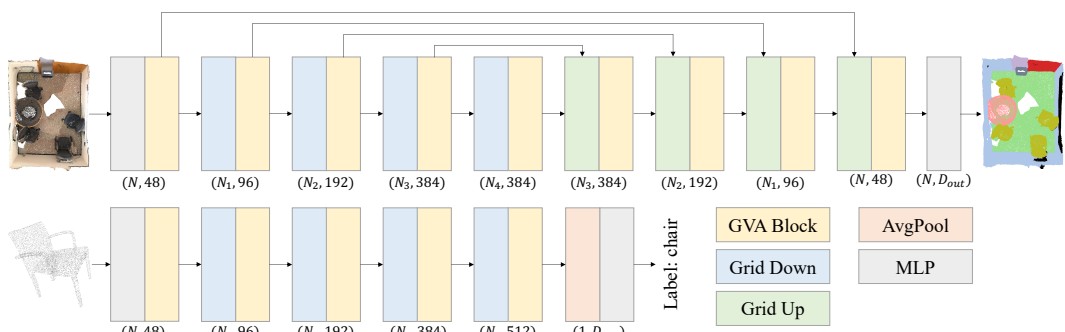

Figure 1: Network architectures for semantic segmentation (top) and classification (bottom).

### A.2  Experiment Setting

**Experimental environment.** Software and hardware environment:

- CUDA version: 11.1
- cuDNN version: 8.0.5
- PyTorch version: 1.10.1
- GPU: Nvidia RTX A6000 $\times$ 4
- CPU: Intel Xeon Platinum 8180 @ 2.50 GHz $\times$ 2

---

*Corresponding Author

36th Conference on Neural Information Processing Systems (NeurIPS 2022).

Table 1: Data augmentation.

| | Drop Points | Rotate | Flip | Scale | Jitter | Distort | Chromatic | Grid Size |
|---|---|---|---|---|---|---|---|---|
| ScanNet v2 | ✓ | ✓ | ✓ | ✓ | ✓ | ✓ | ✓ | 0.02m |
| S3DIS | ✓ | | ✓ | ✓ | ✓ | | ✓ | 0.05m |
| ModelNet40 | ✓ | | | ✓ | | | | |

Table 2: Training Setting.

| | Epoch | Learning Rate | Weight Decay | Scheduler | Optimizer | Batch Size |
|---|---|---|---|---|---|---|
| ScanNet v2 | 600 | 0.005 | 0.02 | Cosine | AdamW | 16 |
| S3DIS | 3000 | 0.005 | 0.05 | MultiStep | AdamW | 16 |
| ModelNet40 | 300 | 0.05 | 0.0001 | MultiStep | SGD | 32 |

**Data license.** Our experiments use open-source datasets widely applied for 3D recognition research. The ScanNet v2 [1] dataset is under the MIT license, while S3DIS [2] and ModelNet40 [3] have custom licenses that only allow academic use.

**Data preprocessing and augmentation.** For S3DIS and ModelNet40 datasets, we adopt the data preprocessing of PTv1 [4] with slight adjustments to the augmentation. For ScanNet v2, we estimate normal vectors for points as additional feature input. The data augmentation strategies are different for each dataset, as shown in Table 1. The detailed settings for each type of data augmentation will be available in our open-source code.

**Training details.** Our specific model training settings are available in Table 2. For the segmentation task, AdamW is used to reduce overfitting of the model. Scheduler with a cosine annealing policy has a better performance on ScanNet v2, which has more data than S3DIS. We use cross-entropy loss for all experiments.

# B    Additional Quantitative Results

In this section, more quantitative results are provided to validate and analyze our proposed network architecture.

## B.1    Depths of Decoder Blocks

We show ablation experiments on the depths of each decoder block with different unpooling methods in Table 3. Applying at least one attention block in each decoding stage can significantly improve the model performance, and this phenomenon is more evident while utilizing our mapping pooling. But deeper decoders (from a depth of 1 to 2 for each decoder block) do not improve performance. These phenomena are reasonable since the naive unpooling methods such as interpolation and mapping require a learnable block to optimize the sampled features. In 2D, deconvolution with stride is widely used to unpooling and optimize features simultaneously, and such a process usually does not require a deep network.

Table 3: Results of different decoder depths for two upsampling strategy (mIoU%).

| Decoder Depths | [0, 0, 0, 0] | [1, 1, 1, 1] | [2, 2, 2, 2] |
|---|---|---|---|
| Grid Mapping | 73.5 | **75.4** | 74.8 |
| Neighbors Interpolation | 73.6 | 74.4 | 73.9 |

## B.2    Ablation study on Position Encoding Multiplier

Table 4 shows an additional ablation study on PE Multiplier. PE Multiplier does not work well with PTv1 since PTv1 already overfits to the training set. Adding more capacity to the PTv1 will not help improve the performance. On PTv2, the group vector attention (GVA) has an effect of reducing overfitting and enhancing generalization. With GVA restricting the capacity of the attention mechanism, the addition of PE Multiplier can focus on learning complex point cloud positional relations. PE Multiplier compliments group vector attention to achieve a good balance of network capacity.

Table 4: Position Encoding Multiplier ablation.

| PE Multiplier | PTv1 | | PTv2 | |
|---|---|---|---|---|
| | ✗ | ✓ | ✗ | ✓ |
| Params (M) | 11.4 | 14.6 | 9.6 | 12.8 |
| Time (ms) | 1023 | 1055 | 220 | 266 |
| mIoU (%) | 72.3 | 72.1 | 75.0 | 75.4 |

Table 5: Combined pooing and unpooling time comparison.

(a) FPS-kNN pooling and unpooling [5, 4] execution time (ms).

| | n=10K | n=20K | n=40K | n=80K | n=160K | n=320K |
|---|---|---|---|---|---|---|
| r=1/2 | 36 | 91 | 255 | 955 | 3644 | 14336 |
| r=1/4 | 20 | 39 | 131 | 488 | 1842 | 7207 |
| r=1/6 | 15 | 28 | 90 | 334 | 1241 | 4811 |
| r=1/8 | 12 | 22 | 70 | 256 | 942 | 3632 |

(b) Grid-kNN pooling and unpooling [6] execution time (ms).

| | n=10K | n=20K | n=40K | n=80K | n=160K | n=320K |
|---|---|---|---|---|---|---|
| r=1/2 | 4 | 6 | 10 | 17 | 52 | 141 |
| r=1/4 | 4 | 6 | 9 | 17 | 36 | 109 |
| r=1/6 | 4 | 5 | 9 | 16 | 23 | 69 |
| r=1/8 | 4 | 6 | 9 | 16 | 22 | 67 |

(c) Grid pooling and unpooling (ours) execution time (ms).

| | n=10K | n=20K | n=40K | n=80K | n=160K | n=320K |
|---|---|---|---|---|---|---|
| r=1/2 | 0.94 | 0.98 | 1.01 | 1.02 | 1.03 | 1.35 |
| r=1/4 | 0.93 | 0.97 | 0.98 | 0.98 | 1.00 | 1.33 |
| r=1/6 | 0.93 | 0.96 | 0.97 | 0.98 | 1.00 | 1.32 |
| r=1/8 | 0.93 | 0.96 | 1.03 | 0.98 | 1.00 | 1.32 |

## B.3   Comparison of Pooling Methods

We compare three different feature-level pooling methods: FPS-kNN, Grid-kNN, and our partition-based pooling implemented by grid. *FPS-kNN* pooling [5, 4] uses the farthest point sampling (FPS) to sample a specified number of points and then query $k$ nearest neighbor points for pooling. Strided KPConv [6] uses a uniform grid to sample points and then applies the kNN method to index neighbors, leading to an uncontrollable overlapping of the pooling receptive field. We name this method *Grid-kNN Pooling*. Our method values non-overlapping receptive fields and fusion points within each non-overlaping grid cell. To distinguish it from the previous sampling-based method, we name it *Grid Pooling*.

**Benchmark with synthetic data.** Table 5 provides benchmark results for the different pooling methods with synthetic data. We generate $n$ points uniformly at random in the unit cube space. The sampling ratio $r$ the sample size divided by the population size. For comparison, we keep the sampling ratio $r$ the same for different pooling methods. For Grid-kNN and Grid Pooling, the grid size is computed as $(n \times r)^{-\frac{1}{3}}$ since the point cloud is sampled uniformly at random. We show the *combined* time of pooling and uppooling for these three methods.