# OpenReview forum: "Point Transformer V2: Grouped Vector Attention and Partition-based Pooling"
_NeurIPS.cc/2022/Conference — NeurIPS 2022 Accept_

### Official Review · Reviewer_N2zu · 2022-07-01

**Rating:** 6
**Confidence:** 4
**Soundness:** 3 good
**Presentation:** 3 good
**Contribution:** 3 good

**Summary:**

Summary:

---
- Point Transformer V2 (PTv2) proposes a new attention mechanism called group vector attention (GVA) that is more efficient than its multi-head self-attention (MHSA) and vector attention of Point Transformer (PTv1) [[1]] counterparts. The effectiveness of GVA is then tested on the tasks of 3D semantic segmentation and 3D shape classification where it achieves competitive performance.

- PTv2 proposes a novel positional encoding scheme that introduces a multiplier $\delta_{mul}$ to the relation vector in addition to the bias that was introduced in PTv1 $\delta_{bias}$.

- PTv2 proposes to introduce grid fusion downsampling and grid mapping upsampling as more effective, efficient, and more spatially aligned alternative to furthest point downsampling and trilinear interpolation upsampling used in PTv1.

[1]: https://openaccess.thecvf.com/content/ICCV2021/papers/Zhao_Point_Transformer_ICCV_2021_paper.pdf


**Questions:**

Questions and Suggestions:

---
- Is there a particular reason why PTv2 does not compare against Stratified Transformer[[2]] but does compare to another CVPR2022 work titled _"Fast Point Transformer"_[[5]]? [[2]] shows that a vanilla MHSA is effective for 3D discriminative tasks and demonstrates the benefits of using shifted-grid locality type for long range contextual information aggregation.

- I would like to see an ablation where the PTv1 architecture is fixed and only the vector attention is replaced with the proposed GVA. I think this is a more fair comparison and a real test for the effectiveness of the proposed GVA.

- I would like to see a comparison of the final points/second throughput against PTv1 and Stratified Transformer.

- I would like a clarification from the authors on the novelty of the used downsampling and upsampling grid based methods.

I am willing to change my rating if my questions and concerns were addressed reasonably in the rebuttal.

[2]: https://openaccess.thecvf.com/content/CVPR2022/papers/Lai_Stratified_Transformer_for_3D_Point_Cloud_Segmentation_CVPR_2022_paper.pdf

[5]: https://openaccess.thecvf.com/content/CVPR2022/papers/Park_Fast_Point_Transformer_CVPR_2022_paper.pdf

**Limitations:**

Even though it is not critical, the paper does not include shape-part segmentation experiments on ShapeNet part or PartNet.

**Strengths And Weaknesses:**

Strengths:

---

- The proposed GVA is more efficient than the vector attention proposed in PTv1 and MHSA while still maintaining competitive performance on semantic segmentation benchmarks such as S3DIS Area 5 and ScanNet test set and shape classification on ModelNet40.

- The paper provides a thorough ablation study most important of which are attention type (GVA vs MHSA) and neighborhood (local vs shifted-grid). Table 7 highlights the effectiveness of the proposed GVA and positional encoding multiplier.

Weaknesses:

---

- PTv2 does not compare to CVPR2022 work titled _“Stratified Transformer for 3D Point Cloud Segmentation”_ [[2]] that is currently the state-of-the-art point-based method on S3DIS Area 5 benchmark with 72 mIoU and reaches 74.7 mIoU on ScanNet test set as per the ScanNet benchmark website (however paper reports 73.7 mIoU). The work uses vanilla multi-head self-attention (MHSA) and uses shifted grid to aggregate information beyond KNN local neighborhood. [[2]] also introduces positional encoding multiplier but in a much more involved scheme. Having said that, PTv2 does include an interesting ablation study in section 4.3 on the attention type used (MHSA vs GVA) and locality type (Shifted-Grid vs Neighborhood) that supports the authors’ choices.

- The comparison with PTv1, in its current form, is not fair because of the architectural differences. The PTv2 uses an encoder with block depths [2,2,6,2] while PTv1 uses [1,1,1,1]. I think a better assessment of the effectiveness of the proposed GVA would be to fix the PTv1 encoder-decoder architecture (including block depths, positional encoding, embedding dimensions, and downsampling/upsampling methods), and just replace the vector attention operator with the proposed GVA.

- The paper claims that the proposed GVA is more efficient. It is evident from the mathematical elaboration in section 3.2 and Table 4 of supplementary material that GVA is more efficient than vector attention of PTv1 and MHSA. However, I would like to see the final points/second throughput of the network as compared with PTv1 and Stratified Transformer (comparison with Stratified Transformer might be tricky because they use CUDA kernels for their attention mechanism).

- The downsampling and upsampling grid based methods are not novel. They were first introduced in KPConv[[3]] and used by later works such as CloserLook3D[[4]]. Maybe the authors meant to say that they introduced these sampling methods to PTv1 that uses furthest point sampling for downsampling, and trilinear interpolation for upsampling.

[2]: https://openaccess.thecvf.com/content/CVPR2022/papers/Lai_Stratified_Transformer_for_3D_Point_Cloud_Segmentation_CVPR_2022_paper.pdf

[3]: https://openaccess.thecvf.com/content_ICCV_2019/papers/Thomas_KPConv_Flexible_and_Deformable_Convolution_for_Point_Clouds_ICCV_2019_paper.pdf

[4]: https://www.ecva.net/papers/eccv_2020/papers_ECCV/papers/123680324.pdf

---

> ### Author Response · Authors · 2022-08-02
> **Response to Reviewer N2zu**
>
> Thank you for your detailed review and comments. We value your words based on related work and try to address your concerns below and in the revised paper.
>
> ### R1: Overall Comparison with ST
>
> A detailed comparison with Stratified Transformer (ST) is available in the comments below. In summary, compared with ST, PTv2 is superior in terms of *model performance, model size and execution time, and design robustness*.
>
> **(a) Model performance**
>
> In the evaluation protocol of ST, [test-time voting](https://github.com/dvlab-research/Stratified-Transformer/blob/7b2a09218b4362db56a63d065a1e260c8deffd63/test.py#L112-L145) is used by augmenting the input points and voting on the results. However, the results we reported in the paper did not use test-time voting. We show our results with and without test-time voting below to ensure a fair comparison with ST. For all comparisons, we use the results reported in the ST paper. We can see that PTv2 outperforms ST in all scenarios. (SN: ScanNet; S3: S3DIS)
>
> || SN Val mIoU ↑ | SN Val mIoU (voted) ↑| SN Test mIoU ↑ | SN Test mIoU (voted) ↑|
> |-|-|-|-|-|
> |ST|N/A|74.3|N/A|73.7|
> |PTv2|**75.4**|**76.1**|**73.9**|**75.2***|
>
> ||S3 mIoU (voted) ↑|S3 mACC (voted) ↑|S3 OA (voted) ↑|
> |-|-|-|-|
> |ST|72.0|78.1|91.5|
> |PTv2|**72.5**|**78.6**|**91.7**|
>
> *: This result is available on ScanNet benchmark website.
>
> **(b) Model size and execution time**
>
> We report the parameter number, amortized forward time, and training time similar to the settings described in supplementary section B.2. PTv2 is ~1.5x smaller than ST in the number of parameters. PTv2 is ~14x faster than ST for each forward pass, and the training time is ~4.86x shorter than ST. The slow performance of ST is caused by GPU thread blocking when multiple threads access data in the same buffer index simultaneously in its CUDA implementation.
>
> ||Params (M) ↓|Forward Time (ms) ↓|Training Time (h) ↓|
> |-|-|-|-|
> |ST|18.6|3,584|175|
> |PTv1|11.4|1,023|42|
> |PTv2|12.8|256|36|
>
> (c) **Design universality.** PTv2's design is more universal than ST, as PTv2 uses the same hyperparameters in all experiments, for patch encoding, number of stages, and depths. However, ST requires manually tunning for these hyperparameters for different experiments.
>
> ### R2: Clarification of PTv1 Setting in Ablation Study
>
> We agree with this comment. Actually, the original ablation experiments about PTv1 were conducted with the same architecture setting as PTv2.
>
> In our ablation study in Table 7, Exp I is PTv1 with "improved structural adjustments and better data processing" (L 294). Here, we modify PTv1 to have encoder block depths [2,2,6,2], consistent with PTv2's design. Exp II in table 7 keeps everything the same as Exp I, but "just replace the vector attention operator with the proposed GVA". Our model complexity and forward time study report that "the baseline PTv1 is set to be the same depth and feature dimensions as our model architecture for comparison" (Supp L 45).
>
> We acknowledge that we can improve the description and have updated the experiment description in the paper accordingly.
>
> ### R3: Speed Comparison with PTv1 and ST
>
> See response R1 (b).
>
> ### R4: Difference with Strided KPConv and Further Experiments
>
> Although both were inspired by strided convolution, our sampling method differs fundamentally from strided KPConv. Based on Fig. 9 in the KPConv paper and its [data processing code](https://github.com/HuguesTHOMAS/KPConv-PyTorch/blob/1defcd75cf7c0399704a6a9f63d3a550bfb8c1c9/datasets/common.py#L403-L413), Strided KPConv first uses uniform grids to sample the points and then uses kNN to query the neighboring points for further pooling. We call this method **Grid-kNN** sampling. Our approach limits the receptive field of the pooling aggregation to each non-overlapping enclosed voxels, and here we name it **Enclosed-Grid** sampling.
>
> We found that non-overlapping receptive field in pooling improves the model's generalization ability. Our enclosed-grid sampling method is a straightforward implementation to achieve this. Additional experiments on speed and model performance have been added in the supplementary material B.4, proving that our method is simple, fast, and effective.
>
> ### R5: Shapenet Part Segmentation
>
> ShapeNetPart was used in PTv1. However, we believe that the ShapeNetPart dataset has been severely overfitted as the instance average IoU gap between the top 10 point cloud models is less than 0.7% (86.5% ~ 87.2%). Considering the page limitations, we chose not to include the ShapeNetPart dataset in the main paper. We test the performance of PTv2 on Shapenet Part Segmentation without any architecture modification and hyperparameter adjustment, and it can achieve 87.1% instance average IoU and 74.3% class average IoU, compared with PTv1, the improvement is 0.5% and 0.6% respectively.

---

> > ### Comment · Reviewer_N2zu · 2022-08-08
> > **Response to the Rebuttal**
> >
> > I would like to thank the authors for the detailed rebuttal. My questions and concerns were well addressed. I do not have further questions. I'm raising my score to 6.

---

> > > ### Author Response · Authors · 2022-08-09
> > > **Re: Response to the Rebuttal**
> > >
> > > Thanks for your comment and acknowledgment. We will optimize our final paper based on our discussion. A well-designed codebase for point cloud representation learning will also be released with our paper. We believe our work will contribute to the development of our area.

---

### Official Review · Reviewer_EpHU · 2022-07-11

**Rating:** 6
**Confidence:** 4
**Soundness:** 3 good
**Presentation:** 4 excellent
**Contribution:** 3 good

**Summary:**

The paper propose grouped vector attention (GVA) module, which allows information to be exchanged within and between attention groups through a new weight encoding layer. It enhances the location encoding by adding a multiplier to the relational vector. And replaces the farthest-point-sampling-based downsampling method and the interpolation-based upsampling method used in PTv1 with an efficient grid upsampling method of its own setting.

**Questions:**

1) I hope the authors can add a visual explanation of the results of the grouped vector attention module that they used in the ablation experiments.
2) The proposed Position Encoding Multipler seems reasonable. I want to know the impact of the Multipler on the overall network params, inference cost time and downstream task accuracy compared with the encoding scheme $\delta = \theta(p_i - p_j)$ in PTv1[1].
3) The authors keep emphasizing that their grid upsampling is more efficient than the farthest-point-sampling-based downsampling method and the interpolation-based upsampling method. However, I only see the metric results for the downstream task. And only in \textbf{Appendix B. Table 4}, there are a few quantitative results related to cost time and params (I want to see a more detailed comparison of the grid upsampling and downsampling method with PTv1[1], e.g. different \textbf{grid sizes})
4) I don't see any relevant explanation of network loss in the paper and the supporting material. I hope the authors can give detailed loss settings for their experiments on downstream tasks.

In the current version, there are some issues as well. I look forward to the response by the authors. For now, I would recommend a borderline accept rating for the paper.

[1]Hengshuang Zhao, Li Jiang, Jiaya Jia, Philip Torr, and Vladlen Koltun. Point transformer. In ICCV, 2021.

**Limitations:**

Yes, I don't see any potential negative social impact of this work.

**Strengths And Weaknesses:**

- Originality: The main idea of the proposed approach is to use the transformer. To my limited knowledge about it, it lacks a little novelty, but the originality is reasonable.
- Quality：While the approach seems reasonable and the experimental results look promising, I have the following concerns (See Questions) about the paper.
- Clarity：This manuscript is clearly written.
- Significance：There are not many works in the field of point cloud feature learning by transformer, so I think this paper makes a positive contribution.

---

> ### Author Response · Authors · 2022-08-02
> **Response to Reviewer EpHU**
>
> Thank you for your detailed review and comments. We try to address your comments below and in the revised paper.
>
> ### R0: Novelty and Necessity of 3D Transformer Backbone
>
> We believe a mature transformer framework for 3D point cloud representation learning is not yet available.
> *Point Transformer (PTv1)* has a severe over-fitting problem over the training set. *Stratified Transformer (ST)* is complex and not as robust since it requires network architectural change for different datasets. *Fast Point Transformer (FPT)* improves the speed over PTv1, but its accuracy is not improved. We propose PTv2 to address these issues and provide a *simple, efficient and effective* transformer backbone.
>
> ### R1: Visual Explanation of the Results of the GVA
>
> Thank you for your comments. In the ablation studies (Table 7), we provide mIoU comparisons to show the effectiveness of the group vector attention (GVA) module. We also provide visualization results (Fig 3, 4; Supp Fig 5, 6) comparing PTv2 (with GVA, PE Multiplier, Grid down, and Grid up modules enabled) and the ground truth. We haven't included the visual results for the ablation studies of the modules since we believe that the numerical mIoU values are more representative than manually selecting visualization results for each ablation step.
>
> ### R2: Execution Time Analysis of PEM
>
> Supp Table 4 shows the results of our model complexity and the forward pass execution time. Specifically, the last two columns (experiment ⑤ and ⑥) show the effect of the Position Encoding Multiplier (PE Multiplier).
>
> ||⑤ PTv2 w/o PEM|⑥ PTv2 w PEM|
> |-|-|-|
> |Params (M) ↓| 9.6 |12.8|
> |Forward Time (ms) ↓| 220 | 266 |
> |mIoU (%) ↑| 75.0 | 75.4 |
>
> Experiment ⑤ is exactly PTv2 but with the encoding scheme of PTv1. Experiment ⑥ is the full PTv2 model with the PE Multiplier. We can see that with the introduction of PEM, PTv2's parameters increased from 9.6M to 12.8M, and the forward pass time increased from 220ms to 266s.
>
> ### R3: Analysis of Sampling Methods
>
> Detailed analysis of the sampling methods is critical to our claimed efficiency. Based on your comment, we have added the analysis and experiments of the time consumption of sampling methods in the supplementary material (B.4) and below. We will include these in the final version of our paper.
>
> (a) **Time complexity**
>
> Assume we downsample a point cloud from $n$ points to $m$ points, and $m = O(n)$. In the FPS-based method, kNN search is performed after FPS to search for $k$ neighbors, and we also assume $k$ is a constant.
>
> For FPS-based method, the downsample time is $O(mn + mk\log(n)) = O(n^2)$, and the upsample time is $O(nk\log(m)) = O(n\log(n))$. For our PTv2's grid-based method, the downsample time is $O(n)$, and the upsample time is also $O(n)$. Note that the real-world GPU execution time does not reflect the theoretical time complexity. More importantly, FPS is inherently an interactive algorithm and cannot be parallelized, while gird-based sampling can be parallelized.
>
> (b) **Sampling time comparison**
>
> We provide benchmark results for the different sampling methods with synthetic data. We generate $n$ points uniformly at random in the unit cube space. The sampling ratio $r$ is the sample size divided by the population size. For comparison, we keep the sampling ratio the same for different sampling methods. For our grid-based method, given the number of points $n$ and sampling ratio $r$, the grid size is computed as $(n\times r)^{-\frac{1}{3}}$.
>
> We show the *combined time* of downsampling and upsampling, for the two methods FPS-based (PTv1) and grid-based sampling (ours). Here the number of kNN neighbors is 16, as used in PTv1.
>
> FPS-based (in PTv1) time (ms):
> ||n=10k|n=20k|n=40k|n=80k|n=160k|n=320k|
> |-|-|-|-|-|-|-|
> |r=1/2|36|91|255|955|3644|14336|
> |r=1/4|20|39|131|488|1842|7207|
> |r=1/6|15|28|90|334|1241|4811|
> |r=1/8|12|22|70|256|942|3632|
>
> Grid-based (in ours) time (ms):
> ||n=10k|n=20k|n=40k|n=80k|n=160k|n=320k|
> |-|-|-|-|-|-|-|
> |r=1/2|0.94|0.98|1.01|1.02|1.03|1.35|
> |r=1/4|0.93|0.97|0.98|0.98|1.0|1.33|
> |r=1/6|0.93|0.96|0.97|0.98|1.0|1.32|
> |r=1/8|0.93|0.96|1.03|0.98|1.0|1.32|
>
> (c) **Forward time and results comparison**
>
> The table below shows the forward time and mIoU of PTv2 with different sampling methods and sampling ratios. Our grid-based sampling method is not only faster but also maintains high mIoU even with a lower sampling ratio ($1/6$).
>
> ||FPS-based|FPS-based|Grid-based (ours)|Grid-based (ours)|
> |-|-|-|-|-|
> |Sample Ratio|1/4|1/6|~1/4|~1/6|
> |Forward Time (ms)|1007|785|318|266|
> |mIoU (%)|74.4|72.9|75.2|75.4|
>
> ### R4: About Network Loss
>
> All our experiments use cross-entropy loss. In Supp Table 2, we only present the differing hyperparameter settings for the three datasets. Based on your suggestion, we have added a relative description to the supplementary material (L28-29).

---

### Official Review · Reviewer_KrWR · 2022-07-11

**Rating:** 5
**Confidence:** 4
**Soundness:** 3 good
**Presentation:** 4 excellent
**Contribution:** 3 good

**Summary:**

This paper upgrades Point Transformer into its version 2 in three ways. First, group vector attention is designed and compared with other alternatives. Second, an additional multiplier is added to the position encoding. Third, a light-weight grid-based sampling is proposed. Point Transformer V2 achieves competitive results in point cloud semantic segmentation and classification tasks.

**Questions:**

Please see the Weaknesses section.


**Limitations:**

No additional limitations or potential negative societal impact.


**Strengths And Weaknesses:**

Strengths:
1. The paper is well-presented with clear illustrations such as Figure 2.
2. The discussion on the  attention mechanism is interesting and insightful. I especially like the degenerate analysis.
3. Solid improvements on fundamental point cloud tasks are obtained.

Weaknesses:
1. It is more complete to include a benchmark for part segmentation.
2. Line 113, I do not think shifted-grid attention is a “common” practice especially the references are Swin Transformer V1 and V2. Besides, the shifted window technique that Swin Transformer employs essentially leads to overlapping windows. Author’s comments on using non-overlapping windows.
3. The investigation in the position encoding multiplier is relatively insufficient, other than one row of ablation experiment. What does it really do?
4. A complexity analysis of FPS vs grid-based sampling will be helpful.

---

> ### Author Response · Authors · 2022-08-02
> **Response to Reviewer KrWR**
>
> Thank you for your detailed review and comments. We try to address your comments below and in the revised paper.
>
> ### R1: About Shapenet Part Segmentation
>
> ShapeNetPart was used in PTv1. However, we believe that the ShapeNetPart dataset has been severely overfitted as the instance average IoU gap among the top 10 point cloud models is less than 0.7% (86.5% ~ 87.2%). Considering page limitations, we chose not to include the ShapeNetPart dataset in the main paper. We tested the performance of PTv2 on Shapenet Part Segmentation without any architecture modification and hyperparameter adjustment. It can achieve 87.1% instance average IoU and 74.3% class average IoU, compared with PTv1, the improvement is 0.5% and 0.6%, respectively.
>
> ### R2: About Shifted-grid Attention
>
> Thank you for the suggestion. Shifted-grid attention was first proposed by Swin, and then became a "common" practice and is widely applied today. We have modified the corresponding description in the paper.
>
> Regarding the description of non-overlapping windows, the Swin Transformer paper mentions "non-overlapping local windows while also allowing for cross-window connection" in its abstract. The Shifted window technique achieves connections between windows to a certain extent but does not change its non-overlapping nature.
>
> ### R3: More Explanation About PEM
>
> Different from the uniform and discrete pixels in images, point cloud coordinates are non-uniform and continuous, resulting in increased complexity of the relative position among points. Therefore, it is crucial to enhance the weight of position relation during relation encoding. We introduce the PE Multiplier acting on QK-relation $r_{qk}$ to achieve this.
>
> The table below shows an additional ablation study on PE Multiplier. PE Multiplier does not work well with PTv1 since PTv1 already overfits the training set. Adding more capacity to the PTv1 will not help improve the performance. On PTv2, the group vector attention (GVA) reduces overfitting and enhances generalization. With GVA restricting the capacity of the attention mechanism, the addition of PE Multiplier can focus on learning complex point cloud positional relations. PE Multiplier compliments group vector attention to achieve a good balance of network capacity.
>
> ||PTv1 w/o PEM|PTv1 w PEM|PTv2 w/o PEM|PTv2 w PEM|
> |-|-|-|-|-|
> |Params (M)|11.4|14.6| 9.6 |12.8|
> |Time (ms)|1023|1055| **220** | 266 |
> |mIoU (%)|72.3|72.1| 75.0 | **75.4** |
>
> We have added the above discussions on PEM in the revised supplementary material (B.3). We will also add it to the final version of the main paper.
>
> ### R4: Complexity and Execution Time Analysis of Sampling Methods
>
> We have added complexity analysis and experiments of sampling methods in the supplementary material (B.4) and below.
>
> (a) **Time complexity**
>
> Assume we downsample a point cloud from $n$ points to $m$ points, where $m = O(n)$. In the FPS-kNN method, kNN search is performed after FPS to search for $k$ neighbors, and we also assume $k$ is a constant.
>
> For FPS-based method, the downsample time is $O(mn + mk\log(n)) = O(n^2)$, and the upsample time is $O(nk\log(m)) = O(n\log(n))$. For our PTv2's grid-based method, the downsample time is $O(n)$, and the upsample time is also $O(n)$. Note that the real-world GPU execution time does not reflect the theoretical time complexity since FPS cannot be parallelized, while our method can be efficiently parallelized.
>
> (b) **Sampling time comparison**
>
> We provide benchmark results for the different sampling methods with synthetic data. We generate $n$ points uniformly at random in the unit cube space. The sampling ratio $r$ equals the sample size divided by the population size. For comparison, we keep the sampling ratio the same for different sampling methods. For our grid-based method, given the number of points $n$ and sampling ratio $r$, the grid size is computed as $(n\times r)^{-\frac{1}{3}}$.
>
> We show the *combined time* of downsampling and upsampling, for the two methods FPS-based (PTv1) and grid-based sampling (ours). Here the number of kNN neighbors is 16, as used in PTv1.
>
> FPS-based (in PTv1) time (ms):
> ||n=10k|n=20k|n=40k|n=80k|n=160k|n=320k|
> |-|-|-|-|-|-|-|
> |r=1/2|36|91|255|955|3644|14336|
> |r=1/4|20|39|131|488|1842|7207|
> |r=1/6|15|28|90|334|1241|4811|
> |r=1/8|12|22|70|256|942|3632|
>
> Grid-based (in ours) time (ms):
> ||n=10k|n=20k|n=40k|n=80k|n=160k|n=320k|
> |-|-|-|-|-|-|-|
> |r=1/2|0.94|0.98|1.01|1.02|1.03|1.35|
> |r=1/4|0.93|0.97|0.98|0.98|1.0|1.33|
> |r=1/6|0.93|0.96|0.97|0.98|1.0|1.32|
> |r=1/8|0.93|0.96|1.03|0.98|1.0|1.32|
>
> (c) **Forward time and results comparison**
>
> The table below shows the forward time and mIoU of PTv2 with different sampling methods and sampling ratios. Our approach is faster and maintains high mIoU even with a lower sampling ratio ($1/6$).
>
> ||FPS-based|FPS-based|Grid-based (ours)|Grid-based (ours)|
> |-|-|-|-|-|
> |Sample Ratio|1/4|1/6|~1/4|~1/6|
> |Forward Time (ms)|1007|785|318|266|
> |mIoU (%)|74.4|72.9|75.2|75.4|

---

### Official Review · Reviewer_MnQR · 2022-07-12

**Rating:** 4
**Confidence:** 4
**Soundness:** 3 good
**Presentation:** 3 good
**Contribution:** 2 fair

**Summary:**

In this paper, the authors proposed Point Transformer V2, which is based on the previous state-of-the-art Point Transformer. Compared to PT V1, PT V2 introduces several improvements, including group vector attention, grouped weight encoding layer, and a novel sampling method. Experimental results on ScanNet V2, S3DIS, and ModelNet40 show demonstrate considerable improvements upon previous methods.

**Questions:**

Please see above weakness.

**Limitations:**

The authors adequately addressed the limitations and potential negative societal impact of their work

**Strengths And Weaknesses:**

Strengths:
I like the presentation and organization of this paper.
Experimental results show significant improvement over previous work.
The grid upsampling/downsampling at the feature level is interesting.

Weakness:
1. One of my concerns is the Group Vector Attention, as presented in Sec. 3.2. From my understanding, it simply applies the group operation to the vector attention in PTV1. I don't think adding a group operation  (or using a group operation to replace MHA) is novel.
The authors claimed, "GVA is a generalized formulation of VA and MSA". This claim is trivial, and we can easily achieve this statement. Besides, when generalizing Group Attention to Multi-head Attention, it can be mathematically generalized, but the implementation may not be easily achieved (we may need pad 0 manually). This generalization is not only suitable for GVA, but is common in all group operations. In Eq. 4, the value should be learnable parameters instead of 1. The Grouped linear is still a simple group operation.

2. In Sec. 3.3, I could not get the motivation of the additional multiplier, which further emphasizes (I would not say enhance) the positional information. Why authors introduce the multiplier to the relation vector? I would like to see a more detailed analysis or theoretical proof, not barren "explanation".

-- For the Sec. 3.4 Grid-based Sampling Methods are more like feature-level voxel or grid sampling. I like this modification, but it is not that novel.

3. About the experiments. The ModelNet40 and S3DIS datasets are relatively simpler, and the performance of models trained on these two datasets varies greatly (maybe because of the overfitting problem?). This is a common issue in the community. Also, ScanNetV2 is not that complex. I would suggest the authors report the mean-std on simple datasets and add more experiments on complex tasks, like 3d object detection.

4. In Table 4, I would like to see an ablation study that uses max-pooling to replace attention, and this study can serve as a baseline.

---

> ### Author Response · Authors · 2022-08-02
> **Response to Reviewer MnQR**
>
> Thank you for your detailed review and comments. We try to address your comments below and in the revised paper.
>
> ## R1
> **(a) Novelty of GVA**
>
> The novelty of the grouped operation is never itself, but its background and motivation. For example, Group Normalization by Kaiming was motivated by group-wise normalization in classical features like SIFT and HOG, which is novel; Grouped Convolution in AlexNet splits convolutions across multiple GPUs to get ground GPU memory limit by grouping is also novel.
>
> PTv1 is structurally different from the classic MSA. PTv2 generalizes these two methods into one unified framework, the GVA. GVA not only introduces grouped operation but also reformulates the attention theory proposed in [SAN](https://arxiv.org/abs/2004.13621) for further generalization to MSA. GVA unifies the SOTA transformer models for point cloud and image. Therefore, we believe our contribution of GVA is novel.
>
> **(b) Generation from GVA to MSA is non-trivial**
>
> Vector attention in PTv1 is structurally different from MSA. Simply adding grouped operations on PTv1 can not lead to an equivalent expression of MSA.
>
> We reorganize vector attention into (1) element-wise relation encoding, (2) vector-wise weight encoding, and (3) value fusion. However, MSA only consists of two steps: qk dot product and value fusion. Our proposal is to decompose qk dot product in MSA into two steps corresponding to (1) and (2), respectively.
>
> Therefore we decompose the dot product (between each pair of query and key vector) into an element-wise product (matching (1)) and a summation of the products (matching (2)). The element-wise product can be viewed as a relation encoding. The summation of products can be viewed as a non-trainable weight encoding function, which we formulated as Eq. 4.
>
> **(c) Correctness of weight encoding function for MSA (Eq. 4)**
>
> The qk dot product in MSA can be decomposed into an element-wise product and a summation of these products. The summation process corresponds to the weight encoding process, which is non-trainable. In MSA, the applied scalar $\frac{1}{\sqrt{c_g}}$ is also not trainable (e.g. in ViT and Swin Transformer). Some variants of MSA applied learnable scalers, but this is beyond the scope of our discussion.
>
> **(d) Using GVA to implement MSA is simple**
>
> To implement MSA with GVA, we only need to replace the *learnable weight encoding layer* in the attention block with the *non-trainable formula* shown in Eq. 4. This only involves changing a few lines of code. We will open source the code along with the paper, including how to implement MSA with GVA.
>
> **(e) Grouped Linear is simple and effective**
>
> The weight encoding of MSA is simple and efficient, while the weight encoding in VA is learnable. The grouped linear generalizes MSA and VA and combines the advantages of these two types of attention mechanisms. The grouped linear alleviates the overfitting problem of PTv1 and is more flexible than MSA in learning accurate patterns of weight encoding.
>
> ## R2
> **(a) PEM**
>
> Different from the aligned and discrete pixels in images, point cloud coordinates are non-uniform and continuous, resulting in increased complexity of the relative position among points. Therefore, it is crucial to enhance the weight of position relation during relation encoding. We introduce the PEM acting on QK-relation $r_{qk}$ to achieve this.
>
> On PTv1, PEM does not work well with PTv1 since PTv1 already overfits the training set. Adding more capacity to the PTv1 will not help improve the performance. On PTv2, the GVA has the effect of reducing overfitting and enhancing generalization. With GVA restricting the capacity of the attention mechanism, the addition of PEM can focus on learning complex point cloud positional relations. Therefore, PEM compliments group vector attention to achieve a good balance of network capacity.
>
> ||PTv1 w/o PEM|PTv1 w PEM|PTv2 w/o PEM|PTv2 w PEM|
> |-|-|-|-|-|
> |Params (M)|11.4|14.6|9.6|12.8|
> |Time (ms)|1023|1055|**220**|266|
> |mIoU (%)|72.3|72.1|75.0|**75.4**|
>
> We have added the above discussions in supp material B.3. We will also add them to the final version of the main paper.
>
> **(b) Sampling**
>
> The novelty is finding the superiority of non-overlapping receptive field in the pooling aggregation step. Grid-based sampling is a straightforward implementation. Additional experiments have been added in the supp material B.4, proving that our method is simple, fast, and effective.
>
> ## R3
> Our model is stable, e.g., 75.26 (0.15) on ScanNet val set (5 repeat). Our result will boost to 76.1 by data augmentation voting in the testing process.
>
> Mean-std results are rare in 3D. 1. Detailed ablation studies cannot afford the resource consumption of repeated experiments for a precise mean-std result. 2. Best results of each proposed model are used for comparison.
>
> ## R4
> PTv2’s baseline is PTv1, while PTv1’s baseline is pooling operator, as you mentioned. The result is available in PTv1 ablation study (Table 7).

---

> > ### Comment · Reviewer_MnQR · 2022-08-08
> > **Response to the authors**
> >
> > Thanks for the authors' response. Here are my responses.
> >
> > 1. Novelty of GVA. I agree that the novelty of the grouped operation is never itself. However, the novelty of GVA is itself since it simply applies grouped operation to vector attention.
> >
> > 2. PTv1 is structurally similar to the classic MSA, but uses different attention implementations. Besides SAN in 2022, [Non-Local in 2018](https://openaccess.thecvf.com/content_cvpr_2018/papers/Wang_Non-Local_Neural_Networks_CVPR_2018_paper.pdf) also introduced several implementations. In the [Transformer paper](https://arxiv.org/abs/1706.03762?context=cs), the authors also claimed that dot-product is not the only implementation.
> >
> > qk dot product is element-wise relation; vector-wise weight encoding (grouped linear layer, followed by batch normalization, activation, and a linear layer) only introduced more layers instead of novelties. Also, the grouped linear has been studied previously in some works, e.g., [ResNeSt](https://github.com/zhanghang1989/ResNeSt/blob/1dfb3e8867e2ece1c28a65c9db1cded2818a2031/resnest/torch/models/splat.py#L46)
> >
> > 3. Thanks for the explanation on the PEM from the perspective of overfitting and generalization.
> > 4. Mean-std results are not rare in 3D, and the computational resource consumption is not unaffordable, especially for the small datasets like modelnet, s3dis, etc. It is crucial to report mean-std results for these small datasets. For example, the results of multiple trainings of this method can vary by more than 1 mIoU on s3dis.
> >
> > 5. Thanks for the clarification.
> >
> > Thanks for the authors' rebuttal again.

---

> > > ### Author Response · Authors · 2022-08-09
> > > **Re: Response to the authors (Part 1)**
> > >
> > > Thanks for the detailed reply to our response. We try to address your concerns with a more precise explanation. Hope our efforts can change your mind.
> > >
> > > **(a) Vector Attention (PTv1, SAN) is a different implement of self-attention, but cannot directly apply multi-head.**
> > > > PTv1 is structurally similar to the classic MSA, but uses different attention implementations.
> > >
> > > In PTv1, dot-product attention (scalar attention) is used as the baseline of vector attention. Scalar attention generates a scalar weight acting on the value vector through the dot product of each pair of query and key. Unlike scalar attention, vector attention directly encodes the element-wise relation between each pair of query and key into vector weight. **Multi-head design can only apply to scalar weight by repeatedly generating independent scalar weights**, and it does not apply to vector attention.
> > >
> > > **(b) Simply applying grouped operation is not our novelty, our novelty is as follows.**
> > > > Novelty of GVA. I agree that the novelty of the grouped operation is never itself. However, the novelty of GVA is itself since it simply applies grouped operation to vector attention.
> > >
> > > 1. **GVA closes the gap between vector attention and multi-head scalar attention from a theoretical framework.** As mentioned in (a), the multi-head design can not be applied in vector attention by simply repeating the generation of independent attention weights. With GVA, we not only prove the superiority of our method from the experimental results but also analyze and compare the properties of the two methods theoretically. Furthermore, GVA provides a theoretical basis for better weight encoding design.
> > > 2. **GVA introduces superior properties to attention weight compared with classic multi-head design.** Attention weights from each attention head are independent in multi-head design. However, attention weights in our GVA are connected, which can make full use of each channel feature of query and key.
> > > 3. **GVA alleviates the overfitting problem of VA.**
> > >
> > > **(c) Different implementations of self-attention do exist, thus an in-depth analysis between vector attention and multi-head scalar (dot-product) attention is crucial.**
> > > > Besides SAN in 2022, Non-Local in 2018 also introduced several implementations. In the Transformer paper, the authors also claimed that dot-product is not the only implementation.
> > >
> > > Multi-head scalar (dot-product) attention is adopted by popular transformer backbones for image understanding (e.g., ViT and Swin). These backbones have been widely applied in several influential works (e.g., MAE and MaskFormer). Our work, the GVA, generalizes vector attention and multi-head scalar attention, and we believe that an in-depth comparison between vector attention and multi-head scalar attention is crucial.
> > >
> > > **(d) qk dot product is not element-wise relation.**
> > > > qk dot product is element-wise relation
> > >
> > > Assume $q \in \mathbb{R}^c$ and $k \in \mathbb{R}^c$ , the output of element-wise relation still belongs to $\mathbb{R}^c$. However, the output of qk dot product is a scalar, which belongs to $\mathbb{R}$. Thus, qk dot product can not be viewed as an element-wise relation. A more detailed explanation is available in previous comment R1(b).
> > >
> > > **(e) Vector-wise weight encoding was proposed by SAN, it is not our design, we optimize the process.**
> > > > vector-wise weight encoding (grouped linear layer, followed by batch normalization, activation, and a linear layer) only introduced more layers instead of novelties
> > >
> > > MLP vector-wise weight encoding was proposed by [SAN](https://arxiv.org/pdf/2004.13621.pdf), which is the theoretical basis of PTv1. Weight encoding was only viewed as "a non-linear mapping" to value channels. In PTv2, we reformulate the vector attention theory and rename it as weight encoding for a better understanding.
> > >
> > > Our novelty can be thought of as **grouped weight encoding** that combines the advantages of vector attention and multi-head scalar attention. This proved to be superior in our experiments and analysis. Our design speeds up the attention scheme in PTv1 instead of introducing more layers to PTv1.

---

> > > ### Author Response · Authors · 2022-08-09
> > > **Re: Response to the authors (Part 2)**
> > >
> > > **(f) About Grouped Linear**
> > > > Also, the grouped linear has been studied previously in some works, e.g., ResNeSt
> > >
> > > 1. Our grouped linear layer (in PTv2) is different from grouped FC (in ResNeSt). The number of output channels of grouped FC is different from the number of groups, while the number of output channels of our grouped linear layer is always equal to the number of groups, as we linearly aggregate elements in each group into a scale value. To some extent, the grouped linear layer can be viewed as a special case of grouped FC.
> > > 2. Grouped linear has a different meaning in our research work. Grouped linear is a learnable weight encoding for multi-head scalar attention. We introduce grouped linear to study the importance of learnable weight encoding without much additional computation compared with multi-head self-attention (only addition backward for learnable parameters).
> > >
> > > **g) Mean-std results.**
> > > > Mean-std results are not rare in 3D, and the computational resource consumption is not unaffordable, especially for the small datasets like modelnet, s3dis, etc. It is crucial to report mean-std results for these small datasets.
> > >
> > > Our mean (std) results of these datasets are as below (SN: ScanNet, S3: S3DIS, MN: ModelNet), our results are better than the results reported in the initial version of our paper due to improved parameter initialization.
> > >
> > > |      | SN Val mIoU ↑ | SN Test mIoU ↑ | S3 mIoU ↑    | S3 OA ↑      | S3 mAcc ↑   | MN OA ↑      | MN mAcc ↑    |
> > > | ---- | ------------- | -------------- | ------------ | ------------ | ----------- | ------------ | ------------ |
> > > |**PTv2** | 75.26 (0.15)  | N/A*           | 72.06 (0.31) | 91.46 (0.20) | 78.2 (0.24) | 93.97 (0.19) | 91.53 (0.22) |
> > >
> > > * Submitting results multiple times on the ScanNet benchmark is not allowed.

---

### Meta-Review · Area_Chair_w4X6 · 2022-08-26

**Recommendation:** Accept
**Confidence:** Certain

**Metareview:**

The paper focuses on modifying specific internal modules in the PointTransformer pipeline for added benefits. It received four detailed reviews from expert reviewers. The discussion period included healthy back-and-forth between the authors and the reviewers, and many of their concerns were addressed and their questions answered. The AC does partially agree with Reviewer MnQR's assessment of the broad impact of this paper, specifically that it may not have a significant influence on the future of research in 3D recognition given the simple technical improvements it proposes. However, the empirical influence of these improvements and their justification/motivation are clear, as clearly stated by the other reviewers. As such, the AC sees that the current level of impact and contribution of the paper reaches the level expected for NeurIPS.

**Award:**

No

---

### Decision · Program_Chairs · 2022-09-14

Accept